# Magnetic slippery extreme icephobic surfaces

Peyman Irajizad[1], Munib Hasnain[1], Nazanin Farokhnia[1], Seyed Mohammad Sajadi[1] & Hadi Ghasemi[1]

Anti-icing surfaces have a critical footprint on daily lives of humans ranging from transportation systems and infrastructure to energy systems, but creation of these surfaces for low temperatures remains elusive. Non-wetting surfaces and liquid-infused surfaces have inspired routes for the development of icephobic surfaces. However, high freezing temperature, high ice adhesion strength, and high cost have restricted their practical applications. Here we report new magnetic slippery surfaces outperforming state-of-the-art icephobic surfaces with a ice formation temperature of $-34\,°C$, 2–3 orders of magnitude higher delay time in ice formation, extremely low ice adhesion strength ($\approx 2\,Pa$) and stability in shear flows up to Reynolds number of $10^5$. In these surfaces, we exploit the magnetic volumetric force to exclude the role of solid–liquid interface in ice formation. We show that these inexpensive surfaces are universal and can be applied to all types of solids (no required micro/nano structuring) with no compromise to their unprecedented properties.

[1] Department of Mechanical Engineering, University of Houston, 4726 Calhoun Road, Houston, Texas 77204-4006, USA. Correspondence and requests for materials should be addressed to H.G. (email: hghasemi@uh.edu).

cephobic surfaces has a crucial role in a wide spectrum of systems including transportation[1–3], infrastructure[4,5] and energy systems[6,7]. However, development of these surfaces for low temperatures is an ongoing challenge. Superhydrophobic surfaces, with an ability to trap air and prevent wetting, minimize the effective contact area of a supercooled droplet and a cold solid substrate to suppress ice formation[8–19]. However, the intrinsic limitation of these surfaces is the existence of a high-energy solid–liquid interface. This interface promotes heterogeneous ice nucleation and enhances adhesion strength of the formed ice on these surfaces. Recently, new icephobic surfaces, slippery liquid-infused porous surfaces (SLIPS)[20–22], were created, which utilize the smooth nature of the liquid surface for icephobicity. The liquid nature of SLIPS boosts mobility of droplets and lowers adhesion strength of the ice–solid interface. However, these surfaces do not show longevity due to depletion of infused-liquid in several icing-deicing cycles[23,24] and are not stable under high shear flows. Furthermore, the same intrinsic limitation, existence of a solid–liquid interface in ice nucleation occurs in SLIPS resulting in a similar freezing temperature and ice adhesion strength as that of superhydrophobic surfaces.

Here we report a new paradigm and the corresponding surface with exceptional icephobicity, high mobility for liquid and ice, self-healing and stability at high Reynolds numbers. In this new approach, magnetic liquid–liquid interfaces are exploited to achieve these unprecedented characteristics. These liquid–liquid interfaces provide a low-energy interface for heterogeneous ice nucleation with Gibbs energy barrier close to the homogenous limit.

## Results

**Magnetic slippery surfaces**. Liquid surfaces are intrinsically smooth and defect-free down to the molecular scale. We used magnetic fluids[25,26] (that is, ferrofluids) along with a magnetic field to develop magnetic slippery surfaces (MAGSS). The characteristics of the ferrofluid is discussed in the Methods section and Supplementary Notes 1 and 2. The use of ferrofluids was based on the premise that it provides a magnetic volumetric force once exposed to a magnetic field; ferrofluid is self-healing in the presence of a magnetic field; the magnetic field locks the ferrofluid in place, which allows for MAGSS to withstand extreme shear stresses; oil-based ferrofluids have a very low evaporation rate allowing for longevity and ferrofluids can be applied to a wide range of surfaces with no required micro/nano fabrication, thereby lowering production costs. The magnetic slippery surfaces are shown schematically in Fig. 1 and are conceptually compared with other state-of-the-art icephobic surfaces. In the classical nucleation theory[27], ice nucleation phenomenon is a competition between the volumetric phase-change Gibbs energy and the surface energy of the involved interfaces. High energy interfaces lower the energy barrier for heterogenous ice nucleation leading to a high freezing temperature. For superhydrophobic surfaces, once a water droplet touches the surface, it forms a high-energy solid–liquid interface with a contact angle of ≥150°. Although the contact area is minimized in these surfaces, the high-energy interface promotes heterogeneous ice nucleation. In the case of SLIPS, once a water droplet contacts the surface, the competition between the interfacial energy of the infused liquid–water interface and the water–air interface leads to infusion of water droplet into the bulk liquid and subsequent formation of a solid–water interface, which is shown in Fig. 1b (see Supplementary Note 3, Supplementary Figs 5 and 6). Formation of this interface (even partial), promotes heterogenous ice nucleation. High ice adhesion strength on SLIPS is another indication of the formation of the solid–liquid

interface. Infusion of a water droplet into a liquid oil bath was studied by Phan et al.[28,29]. The studied oil had lower density than that of water. They found that the downward gravitational force (induced by density difference) can be balanced by interfacial tension forces for certain combination of low surface tension water and oil. However, there was no solid surface in their study and thickness of the oil bath was infinite. We studied the infusion of a water droplet into a finite liquid oil layer through a Hele-Shaw cell in Fig. 1c. In a Hele-Shaw cell with a thickness of 3 mm between two glass slides, we placed a water droplet with a volume of 30 μl on top of a black oil surface, which is in contact with a cold solid substrate at a temperature of −25 °C. The black oil is intentionally used to visualize motion of the water droplet and establishment of a solid–water interface. This method allows us to observe the water–oil interface. Although the buoyancy force opposes infusion of the water droplet to the bulk oil, higher surface energy of the water–air interface compared with the oil–water interface pulls the water droplet to the bulk oil and forms a solid–water interface. Formation of this interface decreases the energy barrier of water–ice phase change and favours heterogeneous ice nucleation. The short ice nucleation time is an indication of the formation of a solid–liquid interface. However, in MAGSS the induced volumetric magnetic force suppresses the formation of a solid–water interface and eliminates the role of this interface in ice nucleation. These characteristics of MAGSS are shown in Fig. 1c, where displacement of the center plane of a water droplet in a ferrofluid medium is examined as a function of the imposed magnetic field. The ferrofluid is contained in a Hele-Shaw configuration and a water droplet is introduced in the ferrofluid. The induced volumetric force by the magnetic field moves the center plane of the water droplet (Supplementary Movie 1, Supplementary Fig. 7). Note that the amplitude of the induced surface waves at the water–ferrofluid interface is a function of the surface tension of this interface and the imposed magnetic field. With low surface tension ferrofluids or high magnetic fields, this amplitude can be decreased (Supplementary Fig. 4).

**Icephobicity of MAGSS**. We develop an experimental set-up to examine icephobicity of MAGSS (Supplementary Note 4, Supplementary Figs 8 and 9). The icephobic surfaces are characterized by three figures of merit: median ice nucleation temperature, $T_N$; average ice nucleation delay time, $\tau_{av}$[30] and ice adhesion strength on the surface. $T_N$ is defined as the ice nucleation temperature of a sessile water droplet placed on a surface when the system of droplet, surface and surrounding is cooled with a slow and quasi-equilibrium approach. $\tau_{av}$ is defined as the average time required for ice nucleation of a supercooled droplet in thermal equilibrium with its surrounding. The $T_N$ of MAGSS is compared with the other state-of-the-art icephobic surfaces in Fig. 2a. Superhydrophobic surfaces and SLIPS offer a $T_N$ range of −23 to −26 °C, while MAGSS offer a $T_N$ value of −34 ± 1 °C (Supplementary Fig. 10). The unprecedented icephobicity of MAGSS is achieved through formation of a low-energy magnetic liquid–liquid interface. In other state-of-the-art surfaces, existence of a solid–water interface limits their icephobicity. We should emphasize that homogeneous limit of ice nucleation in bulk water is −40 °C (refs 31,32). Also, we studied icephobicity of MAGSS under heating/cooling cycles (Supplementary Note 5, Supplementary Fig. 17) and no change in the icephobic characteristics of MAGSS was observed during cyclic performance. The ice nucleation delay time of MAGSS is compared with other icephobic surfaces in Fig. 2b (Supplementary Figs 11–13, Supplementary Table 2). MAGSS outperforms the other surfaces and the ice nucleation time is 2–3

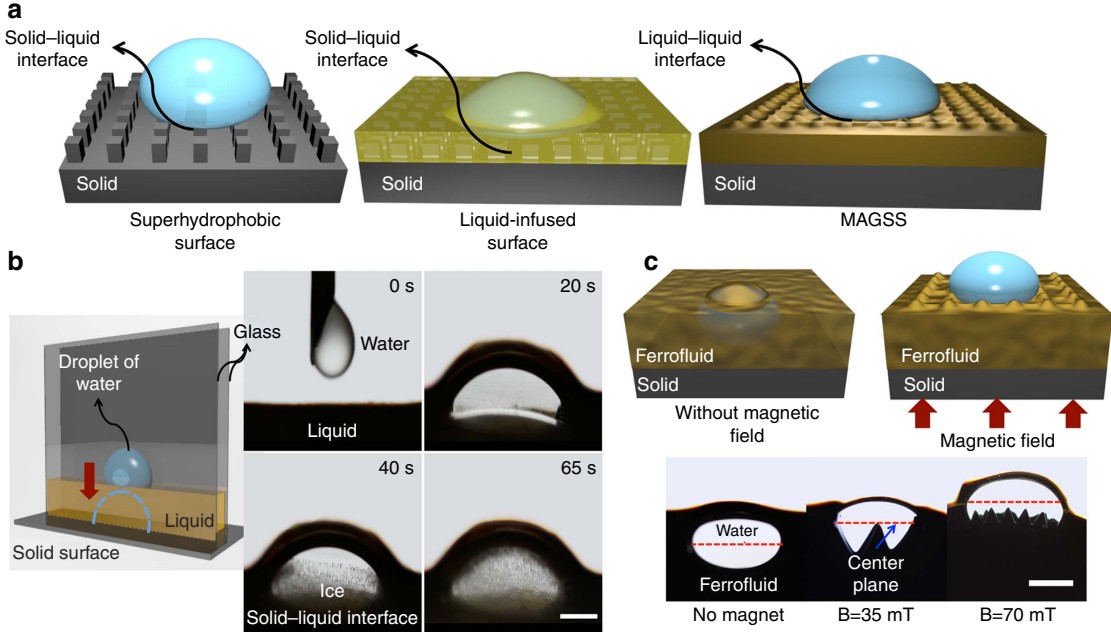

**Figure 1 | Magnetic liquid–liquid interfaces in MAGSS.** Physics of the formation of a liquid–liquid interface in MAGSS is discussed. (**a**) Once a water droplet touches a superhydrophobic surface or a liquid-infused surface, it forms a solid–water interface promoting heterogeneous ice nucleation on these surfaces. However, in MAGSS, the water droplet forms a magnetic liquid–water interface with lower interfacial energy than in the solid–water interfaces. That is, the energy barrier for ice nucleation is increased, and solid temperature should be lowered to satisfy thermodynamic conditions for ice nucleation. (**b**) A Hele-Shaw cell (left side) is used to elucidate penetration of water droplet in the liquid oil layer and formation of solid–water interface, leading to freezing at high temperature, (the kinetics of ice formation is shown on the right side). (**c**) Although the difference in the interfacial energy exists in MAGSS, the induced volumetric force dominates the interfacial force and leads to formation of a magnetic liquid–water interface. In a Hele-Shaw configuration, displacement of the water droplet to the surface in the presence of magnetic field is shown. That is, the volumetric force by the magnetic field does not allow formation of a solid–water interface. Low energy of magnetic liquid–liquid interface increases the energy barrier for ice formation. We emphasize that the amplitude of the induced surface waves at the water-ferrofluid interface can be decreased by low surface tension ferrofluids or higher magnetic fields. (Supplementary Movie 1).

orders of magnitude higher than the other surfaces. In Fig. 2c, the ice nucleation temperature as a function of imposed magnetic field is shown (Supplementary Fig. 14). We noted that $T_N$ is a function of magnetic field, but reaches a threshold value at high magnetic fields. In Supplementary Note 3, we have discussed this threshold of magnetic field. In the presence of a magnetic field, the competition between surface tension force and the magnetic force results in the formation of surface waves with specific periodicity. The water residing in the pits of these surface waves can be in contact with the solid-substrate depending on the magnetic field. This partial existence of a solid–water interface promotes heterogenous ice nucleation as seen for $B \leq 347$ mT. However, once the magnetic field reaches a threshold value, $B = 347$ mT, the solid–water interface is excluded and the ice nucleation time remains approximately constant. With further increase in the magnetic field, we did not observe any change in the ice nucleation temperature and ice nucleation delay time. To avoid partial formation of solid–liquid interface at lower magnetic fields, one may consider increasing thickness of MAGSS. As a comparison, the $T_N$ and $\tau_{av}$ for SLIPS is shown in Supplementary Figs 15 and 16 which the values are consistent with the previous studies[22]. The last figure of merit of icephobic surfaces is shown in Fig. 2d. The ice adhesion strength on MAGSS is measured through the required shear force for sliding of ice on MAGSS surface (see Supplementary Note 5). The low value of shear force is induced by tilting the MAGSS surface after ice formation. The shear strength of ice on MAGSS is $\approx 2$ Pa. This shear strength is five orders of magnitude lower than the reported values for superhydrophobic surfaces and SLIPS surfaces[23,33]. We emphasize that even at extremely low temperatures, once ice

forms on MAGSS, it slides from the surface by a minimal force preventing ice accretion on the surface. The low shear strength is a direct consequence of the magnetic slippery liquid interface. We should add that recently Golovin *et al.*[34] utilized interfacial slippage to achieve icephobic surfaces with ice adhesion strength of 200 Pa. In their approach, they introduced uncross-linked polymeric chains on the surface to achieve no-slip boundary condition at the solid–ice interface.

**Surface characteristics of MAGSS.** Once a water droplet sits on MAGSS, fast removal of the droplet reduces the contact time and the possibility of ice formation. We have examined motion of water droplets on MAGSS at a temperature of $-26\,°C$ in the ambient environment as shown in Fig. 3a. With a tilt angle of 2.5° (Force of 12 µN), the droplet shows high mobility with velocity of $37.5$ mm s$^{-1}$ and a contact angle hysteresis of 1° (Supplementary Movies 2–4). In addition to the mobility of water droplets, the fast removal of the formed ice on MAGSS is required to avoid ice accretion. After formation of ice on MAGSS, we applied a 2.5° tilt to the surface (Force of 11 µN). The ice is immediately removed from the surface with a velocity of $0.8$ mm s$^{-1}$ (Supplementary Movie 5). In addition to Si wafer, we formed MAGSS on glass and acrylic surfaces and observed the same mobility of ice on these solids. That is, with MAGSS, the water or ice does not experience any friction by the underlying solid substrate. This exceptional mobility is a direct consequence of the molecular-scale smoothness of MAGSS and the lack of pinning of a water droplet or ice. In Fig. 3b, we examined the mobility of water droplets on MAGSS for a range of water droplets and tilt angles. Only with

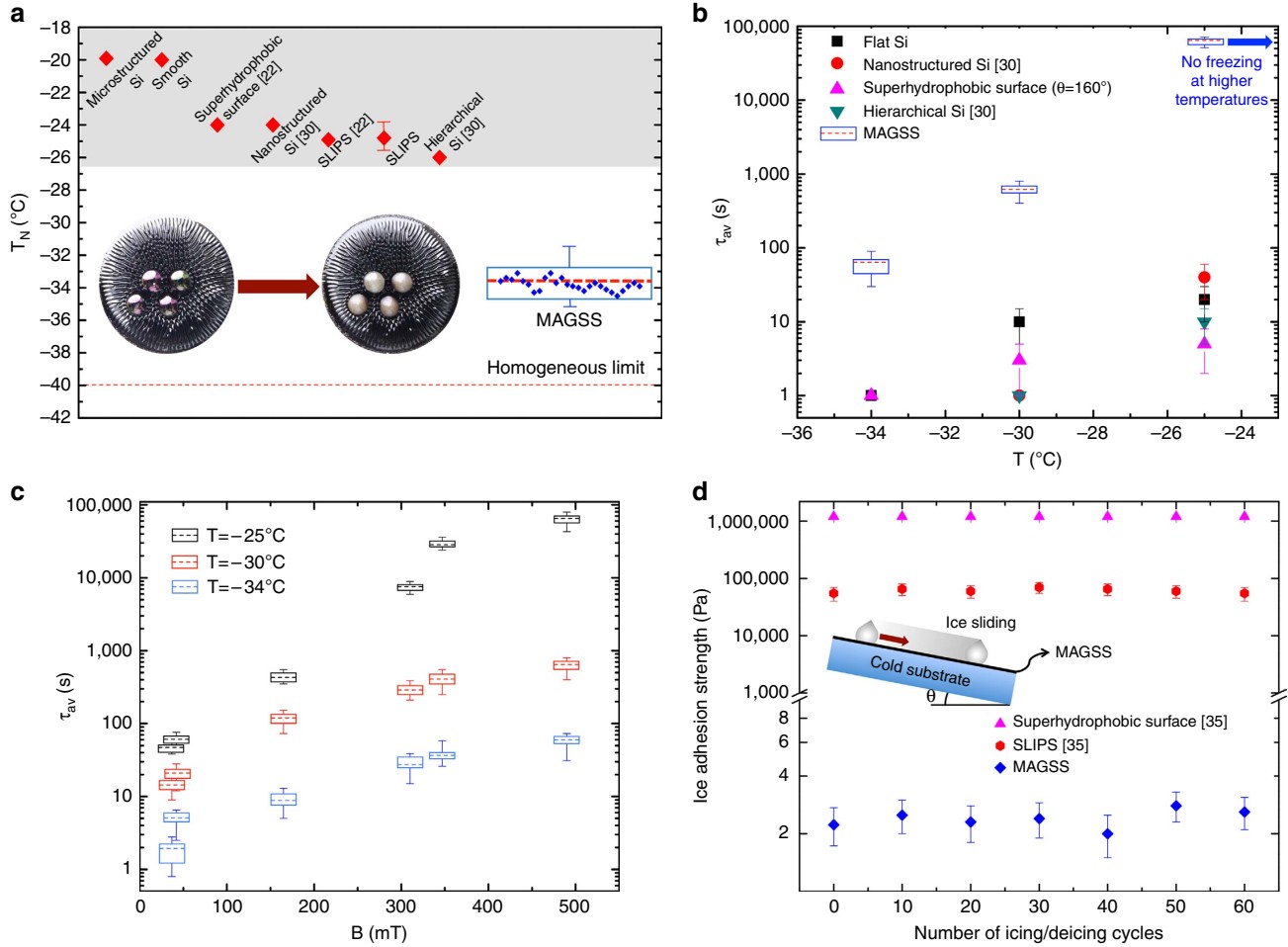

**Figure 2 | Exceptional icephobicity of MAGSS.** Exceptional icephobicity of MAGSS is studied with three figures of merit. (**a**) Median nucleation temperature of MAGSS is compared with other state-of-the-art icephobic surfaces. We achieved a $T_N$ value of $-34 \pm 1$ °C for MAGSS. This value is the median of 35 measurements, which are included as the error bar (see Supplementary Note 5). The formation of a low-energy magnetic liquid–liquid interface limits the ice formation to extremely low temperatures close to homogeneous nucleation limit. The inset shows the structure of MAGSS before and after ice formation. Note that there is essentially no change in the structure of MAGSS after the freezing process. (**b**) Ice nucleation delay time for MAGSS is compared with other icephobic surfaces. MAGSS outperforms the other surfaces over all temperature ranges and the ice nucleation delay time is 2–3 orders of magnitude higher than state-of-the-art icephobic surfaces. Note that at temperatures higher than $-25$ °C, we could not detect any ice nucleation on the MAGSS surfaces after 27 h. (**c**) At lower magnetic fields, $\tau_{av}$ is proportional to the imposed magnetic field, but at higher magnetic fields and complete exclusion of solid–water interface, $\tau_{av}$ remains constant (see Supplementary Note 5). (**d**) The ice adhesion strength on MAGSS is five orders of magnitude lower than the other icephobic surfaces. The exceptional ice adhesion strength is achieved through magnetic slippery liquid–liquid interface. The number of measurements for each figure of merit and their statistics are provided in Supplementary Note 5. The range of measured data are shown as error bars.

tilt angle of $<5°$ could we detect pinning for droplets smaller than 0.3 mm in diameter. For larger droplets, no pinning was observed. As shown, MAGSS shows exceptional mobility compared to other state-of-the-art surfaces[35]. In the next step, we examined self-healing characteristics of MAGSS through water droplet impact experiments. The sequence of images in Fig. 3c show how the surface is deformed by droplet impact and then fully healed in no more than 70 ms (see Supplementary Movies 6 and 7). Furthermore, we examined the self-healing characteristics of MAGGS when scratched by a sharp object as shown in Supplementary Movie 8. MAGSS surface remain unaffected due to scratching and hitting by a sharp object. In some applications of icephobic surfaces such as aircrafts and ocean-going vessels, the icephobic surfaces are imposed to high shear fluid flows. We exposed MAGSS to high-shear nitrogen gas flow (Supplementary Note 5) and examined the stability of the ferrofluid layer. The assumed critical threshold was depletion rate ($\leq 2\,\mu m$ per hour) of ferrofluid from the surface. The critical Reynolds number for

the stability of the ferrofluid is shown in Fig. 3d as a function of the imposed magnetic field. The critical Reynolds number at the imposed magnetic field of 390 mT is $\approx 10^5$ (Supplementary Movie 9). We conducted the stability experiments with water flow up to Reynolds number of 2,000 and could not detect any depletion in the MAGSS (Supplementary Movie 10). We ran these experiments for $>15\,h$ to examine longevity of these surfaces. The magnetic field locks MAGSS in place providing longevity for performance under high shear flows. Furthermore, we studied the role of the size of water droplets on icephobicity of MAGSS. We examined droplets in the range of 4–180 µl and could not detect any measurable change in the value of $T_N$ (Supplementary Note 6, Supplementary Fig. 18).

## Discussion

To elucidate underpinnings of exceptional icephobicity of MAGSS, we studied heterogenous ice nucleation on MAGSS.

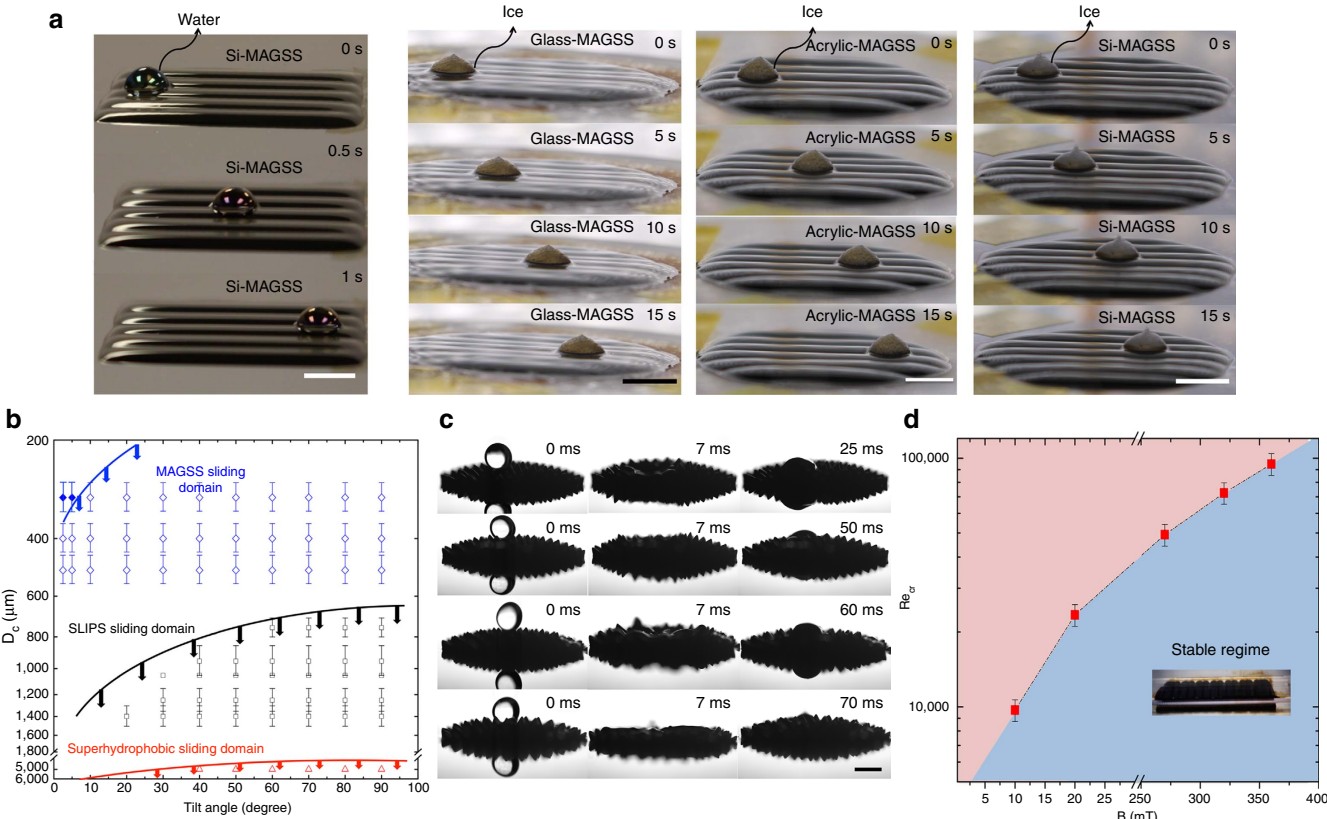

**Figure 3 | Characteristics of MAGSS.** Surface characteristics of MAGSS, self-healing and stability under high-shear flows are studied. (**a**) The high mobility of a water droplet and ice on MAGSS is shown. For a water droplet with tilt angle of 2.5°, the droplet moves with a velocity of 37.5 mm s$^{-1}$ (Supplementary Movies 2–5). For ice with a similar tilt angle, ice has a velocity of 0.8 mm s$^{-1}$. We have shown same mobility of ice with MAGSS developed on glass and acrylic surfaces indicating that mobility is independent of the underlying solid surface. That is, in MAGSS, neither water nor ice experiences any friction by the solid substrate. (**b**) Mobility of the water droplets on MAGSS is examined for different droplet diameters and tilt angles. As shown, MAGSS offers unprecedented mobility compared with other state-of-the-art surfaces. High mobility of droplets on MAGSS reduces the contact time of droplets with the cold surface and hinders possibility of ice nucleation. The error bars represent uncertainty in the experimental approach. (**c**) The self-healing characteristics of MAGSS is assessed through droplet impact experiments. At $t = 0$ ms, the droplet is just about to impact the surface. At $t = 7$ ms, the perturbation of the surface is evident and as the impact velocity is increased, so does the amount of perturbation. The last image in the sequence shows the completely healed MAGSS. MAGSS has revived its structure in $<70$ ms with no sign of splashing or depletion of ferrofluid. The Weber numbers are 18.87, 30.78, 71.82 and 154.68, respectively (Supplementary Movie 6). For higher Weber numbers (600), we used high viscosity ferrofluid and no change in the characteristics of the surface was observed (Supplementary Video 7). (**d**) Stability of MAGSS under an imposed high-Reynolds number shear flow is shown. Critical Reynolds number is a function of the magnetic field. The assumed critical threshold was depletion rate ($\leq 2$ μm per hour) of ferrofluid from the surface. At higher magnetic fields, MAGSS is stable up to a Reynolds number of $10^5$. (Supplementary Movies 9 and 10). We ran these experiments for more than 15 h to examine longevity of these surfaces. Scale bar, 1 mm.

A schematic of a water droplet on MAGSS is shown in Fig. 4a. The energy barrier for ice nucleation is written as[27]

$$\Delta G^* = \frac{8\pi\gamma_{IW}^3}{3\Delta G_v^2} f(m, x), \qquad (1)$$

where $\Delta G^*$ denotes the Gibbs energy barrier, $\gamma_{IW}$ the surface energy of ice–water interface, $\Delta G_v$ the volumetric free energy of phase-change and $f(m, x)$ a geometrical factor (Supplementary Fig. 19). In $f$ function, $m = cos(\theta_{IW})$, in which $\theta_{IW}$ denotes the contact angle of the ice–water interface, and $x$ is the ratio of radius of heterogeneous nucleation site to critical ice nucleus radius. The route to exceptional icephobicity is to tune function $f(m, x)$ to approach unity (for example, Homogenous nucleation limit). Since the magnetic liquid–liquid interface is smooth in molecular scale (see Methods section), $x$ value in MAGSS approaches infinity. We plotted $f(m, x)$ as a function of $x$ for different values of $m$ in Fig. 4b. As shown, at high values of $x$, $f$ function is independent of $x$. That is, in MAGSS, exceptional

icephobicity is a direct consequence of the $m$ value. We have developed an approach to determine the $m$ value in MAGSS surface (Supplementary Note 7, Supplementary Fig. 20, Supplementary Table 1). In these calculations, the surface tension of the ice–water interface was extracted from ref. 36. As shown in Fig. 4c, this value is less than $-0.95$ and is a decreasing function of temperature. This finding suggests that the low surface energy of magnetic liquid–liquid interface shifts the ice nucleation towards the homogeneous nucleation limit. We could not detect any dependence of the $m$ value on the magnitude of the magnetic field. Once we determined the value of $m$, we used nucleation theory to predict the value of average ice nucleation delay time as a function of temperature. These calculations are provided in Supplementary Note 7. We found that the measured values of $\tau_{av}$ agree with these predictions, Supplementary Fig. 21.

The low interfacial energy of magnetic liquid–liquid interfaces opens a new path to develop surfaces with exceptional icephobicity, unprecedented low ice adhesion strength ($\approx 2$ pa), negligible friction to water and ice motion, self-healing

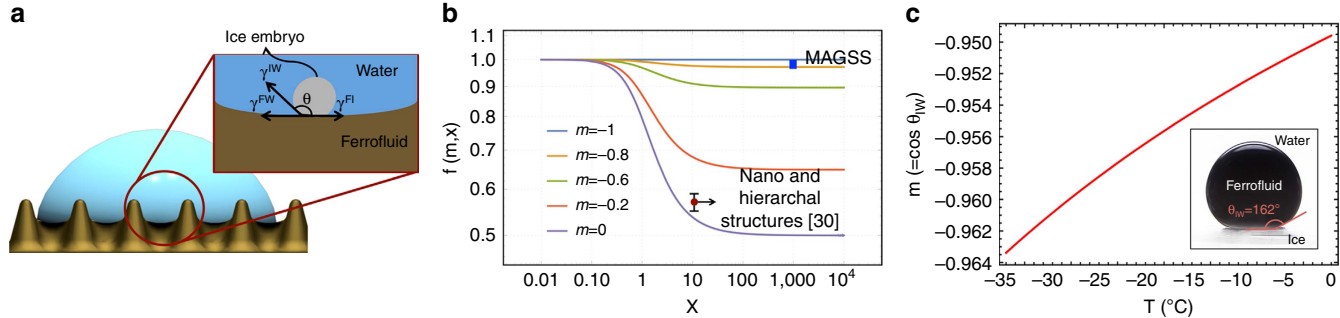

**Figure 4 | Mechanism of exceptional icephobicity in MAGSS.** Physics of the exceptional icephobicity in MAGSS is discussed. (**a**) Schematic of water droplet on MAGSS; an ice embryo at the water–MAGSS interface along with involved interfacial energies are shown. (**b**) The geometrical function of heterogeneous ice nucleation, $f(m, x)$ is plotted as a function of $m$ and $x$. The $f$ value of unity corresponds to the homogeneous nucleation limit and smaller $f$ functions favours heterogenous ice nucleation. The $m$ value for MAGSS and other state-of-the-art hierarchical structures are shown. (**c**) The value of $m$ for MAGSS is measured as a function of temperature. The contact angle of a ferrofluid droplet on ice in a water environment is used to determine the value of $m$ (inset of **c**). The $m$ value of MAGSS is less than $-0.95$, suggesting that MAGSS raises the energy barrier for ice nucleation to close to homogenous nucleation limit.

characteristics and high shear flow stability. These surfaces (MAGSS) provide a defect-free surface for ice nucleation and thereby lower the ice formation to close to homogenous nucleation limit. These surfaces promise a new paradigm for development of icephobic surfaces in aviation technologies, ocean-going vessels, power transmission lines and wind turbines in extreme environments.

## Methods

**Materials.** The ferrofluid used for all experiments was an oil-based ferrofluid from CMS magnetics Inc. (Part Number: FERRO-2OZ). The density of ferrofluid was $1,064\,\mathrm{kg\,m^{-3}}$. This ferrofluid was chosen for its high saturation magnetization. We measured the saturation magnetization of this ferrofluid through the Gouy method[25] as shown in Supplementary Fig. 1. The magnetic field was introduced through neodymium magnets from McMaster-Carr and Eclipse Magnetics. We measured the chemical composition of the ferrofluid through Fourier transform infrared spectroscopy (FTIR) (Nicolet 4,700 FTIR, Thermo Electron Corporation). These measurements are shown in Supplementary Fig. 2. The oil seems to mainly consist of a family of fatty acids, primarily Lauric acic ($CH_3(CH_2)_{10}COOH$). At a wavenumber of $3,000\,\mathrm{cm^{-1}}$, the peak of carboxylic acid group ($-COOH$) appears. The peak corresponding to the alkenyl group ($R\text{-}CH = CH_2$) appears around a wavenumber of $1,100\,\mathrm{cm^{-1}}$. Traces of other fatty acids such as myristic and palmitic acids can also be seen in the FTIR spectrum. The single domain iron-oxide nano-particles in the ferrofluid are coated with surfactants to be suspended in the ferrofluid matrix. The used surfactants are proprietary information of CMS magnetics Inc. and we could not obtain information on the chemical formulas of these surfactants. To determine the dimensions of the iron-oxide nanoparticles, we diluted the ferrofluid with acetone several times. A smooth Si substrate was prepared by plasma cleaning. A droplet of diluted ferrofluid was deposited on the Si substrate and the liquid was allowed to evaporate. A ring trace of the particles was formed on the surface due to coffee-ring effect. The sample was placed in the scanning probe microscope (Bruker Multimedia 8) and the distribution of particle size was measured. We noted that aggregation of particles occurred in the evaporation process. However, the smallest particles were in the range of 10–20 nm as shown in Supplementary Fig. 3. We should emphasize that these nano-particles are covered with surfactant molecules. That is, once they float on the surface of the ferrofluid, the roughness of the ferrofluid surface is determined by the structure of surfactant molecules and consequently can be assumed the molecularly smooth. Unless specified otherwise, the substrate used was a silicon wafer purchased from nova electronic materials, LLC. Distilled water was used for all measurements. In these experiments, a chiller from polyscience (AP15R-40-A11B) with refrigerant R-404A was used to obtain temperatures between $-40$ and $+200\,^\circ C$. The refrigerant was directed to flow through a tube connected to an aluminum cold plate.

**MAGSS preparation.** Each substrate was cleaned in a plasma cleaner (Harrick Plasma, PDC-001-HP) for 1 min before its use in the experiments. The neodymium magnet was placed underneath the substrate in order to secure the ferrofluid that was to be added on top of the substrate. The ferrofluid was added using a syringe. Adding a specific volume to a known surface area allowed us to control the thickness of the ferrofluid layer. The thickness of the ferrofluid in our experiments was 300 µm. The surface area of the ferrofluid on the substrate is the same as the surface area of the top surface of the magnet.

**Data availability.** Data supporting the findings of this study are available within the article and its Supplementary Information files.

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

## Acknowledgements

H.G. acknowledges funding support from University of Houston and Air Force Office of Scientific Research (AFOSR) under contract no. 110941 with Dr Ali Sayir as Program Manager. We thank Varun Kashyap, Sahil Ray and Mohammad S. Safari for their assistance with the experiments.

## Author contributions

H.G. conceived the research. P.I. and M.H. developed the experimental set-up. P.I., M.H., and N.F. conducted the icephobicity experiments. P.I., M.H. and S.M.S. conducted high shear flow, mobility and droplet impact experiments. P.I. and H.G. conducted the theoretical analysis. P.I., M.H., and H.G. wrote the manuscript.

## Additional information

**Competing financial interests:** The authors declare no competing financial interests.

**Publisher's note**: 

