## [Peer Review File · Nature Communications]

Reviewers' comments:

Reviewer #1 (Remarks to the Author):

The idea of icephobicity using magnetic surfaces is both novel and fascinating. the results look encouraging indeed and as such should be published sooner rather than later. The measurement of lag times (time until nucleation for supercooled droplets) is probably not the best way to proceed however and some further measurements are required with multiple measurements on each sample.

I have no doubt the conclusions are valid - but they have not been sufficiently proved yet - it is well known that a given liquid sample if supercooled may freeze after say 200 seconds but on the next run may not freeze for 10000 seconds. This stochasticity seems not to have been discussed in that aspect of the measurements.

Specific comments:

p1,l18 - normally do not have acronyms in abstracts

p2,32,33 the English needs fixing, i.e. longevity due to ...

p2, l38 dont use paradigm twice in same paragraph

p3, l53 delete the before hetero...

p3 l57 - explain better that photograph of the water becoming embedded - it is not clear when compared to slips coatings

p5, l2 more explanation needed of droplet shapes typically on superhydrophobic surfaces

p5 l84 - promotes nucleation - no evidence and not generally what is found at all - delete or explain

p7 fig 2 b, need to know sample numbers for each test - if n=1 then need to be done 10 times as minimum

p7 l 124 - again n=?

p12 l203 delay change to drop or lower

Reviewer #2 (Remarks to the Author):

Review of NCOMMS-16-10411-T "Magnetic extreme icephobic surfaces" by Ghasemi and co-workers

This paper reports an experimental study on a new kind of icephobic surface with extreme performance, which the authors demonstrate with a series of experiments. There are a few things that from the outset seriously limit my enthusiasm about this work. The first is that this is an "active" surface. It requires a magnetic field the magnitude of which depends on application and surface size and as such it is inherently limited to ferrofluids that are responsive to this. Think of an airplane wing, a power transmission line, the surface of a ship or the blade of a large wind turbine with a liquid (ferrofluid) surface that is magnetically activated, as the authors mention at the end of the abstract. It is very difficult (actually impossible) to believe that these applications have in reality any relevance to the lab level surfaces presented in this paper.

Then the authors expend a serious effort trying to claim that their surface is a new paradigm and not the same with SLIPS. I disagree. They rely on the same principle (slipping on a liquid layer), with the additional very serious restriction that their surface needs to be magnetically activated. SLIPS are passive surfaces-a huge advantage in my opinion (and we still have to see any significant applications even with SLIPS). In the original SLIPS papers (the one by Aizenberg and co-workers, for example) actually the droplet was simply riding on a film of liquid (not sinking through it), the porous texture

servicing the stability of the film. In this respect the claim that the magnetic liquid needs not texture (also vis. claimed advantage of no contact with the solid of the supercooled water hence ice nucleation inhibition) is not proven/correct.

Once we allow for fields, electrical, magnetic etc to impart the surface functionality, then it is much easier to reach impressive numbers (as in this paper with very high supercooling and very low adhesion) but then the real problem is that these surfaces remain at the impressive "toy" level with no chance to real applications.

So, I find it a lot more appropriate to say that the MAGSS surfaces of this paper are as special case of a limited active SLIPS surface, rather than they are a new paradigm and all these other superlatives the authors use for their surface (overselling). This is what I think is the real perspective on the importance and novelty of this work.

Some additional comments:

- In Figure 1 the explanations on possible heterogeneous nucleation of the droplet on a SLIP surface are purely speculative and indeed self serving. No experiments whatsoever were performed on correctly fabricated and optimized SLIP surfaces for this purpose and no measurements were made of the corresponding TN and τ_{av} to prove a real comparison, despite the big claims of the authors. Results of other studies for other materials and conditions are shown for comparison, but this is far from conclusive. These other surfaces are all passive and no attempt was made (for example in the SLIP) surfaces to optimize with respect to nucleation.
- Regarding the same measurements with the magnetic surfaces of this paper: Little details are given in the SI. These measurements are tricky and require statistics, usually from observing many drops simultaneously and determining freezing (or condensation, it is similar) probabilities with defined error ranges. None of this is done. Also what is the effect of environmental conditions in the chamber? One has to also monitor humidity to define the thermodynamic state. What is this effect? None of this is done/explained. Note that due to exponential behaviors such measurements are very sensitive. I am not convinced of the correctness of the reported results.
- No thermodynamic analysis has been made to come up with quantitative criteria on surface energies of various configurations and enable a scientific conclusion, other than observation from experiments. This should have been a must for the authors to establish the criteria under which the so-called volumetric magnetic force overcomes the surface energy differences they claim.
- No analysis has been performed to quantify the obviously formed surface waves or non uniformities on the surface of the ferromagnetic liquid, their dependence on the various process parameters and their effect on the entire process. A value of $B=347$ mT is mentioned for the specific experiments performed in this paper after which the solid water interface is excluded and the nucleation time remains constant. This is purely empirical and specific. One needs a thorough general analysis for useful generic conclusions.
- The adhesion test results are impressive but again, this is a magnetically activated effect with little bearing to real applications in technology nor manifestation in nature. Is it at all relevant to compare these results to passive surfaces (Lotus-based, or SLIP?).
- Durability tests and videos are shown for cases showing robustness for about 30 min. In what real application mentioned by the authors (turbines, boats, powerlines and aviation) is this nearly enough? And how are the high magnetic fields to be applied to ensure durability at high Weber, and Reynolds numbers? The reported We numbers for example are low. Finally how about mechanical tests? Resistance to scraping off of the ferrofluid that removes the entire functionality? What fields does this need as a function of the scraping force?
-

In closing, based on the above I cannot recommend this paper for publication. I recommend that the authors reduce their grandiose claims to realistic statements placing the work into proper perspective,

improve their results as I suggest above and then I can see the manuscript as publishable in a specialized journal, although the relevance to application will still be missing.

Reviewer #3 (Remarks to the Author):

Please see the attachment

The paper by Irajizad et al. reports a new way to make icephobic surfaces using magnetically energized ferrofluid on surfaces. It is a creative and simple approach and there are some exciting experimental results which will be of broad interest. I am inclined to support the publication of the work, but there are some major scientific aspects that need to be clarified.

Major comments:

1. The physics of freezing delay on MAGSS is not clear and needs more work. Vital to the working of these *icephobic surfaces* is the stability of a water droplet on the field activated ferrofluid puddle (with Rosensweig projections). Under no magnetic field the water droplet sinks into the ferrofluid, which ruins the icephobicity. Therefore, it is critical to understand this stability. Simply making a broad statement that with magnetic field the body forces will start to dominate the interfacial forces (as the authors do on the page 6 of the manuscript, "In the presence of ...") does not address this. Freezing delay is an interfacial surface energy dependent phenomenon. Why should the change in body force - *in the ferrofluid* – influence stability of a water droplet sitting on it or the freezing of the water droplet?
The authors will need to assess the rheology of the ferrofluid under magnetic field and then use a force balance to help substantiate their results and explain them. Please see the works by "Phan et al. Can Water Float on Oil? *Langmuir* 2012, 28, 4609–4613." It offers insight on interfacial force balance and droplet stability.
2. Looking at the interfacial tension values measured (section 7 of the Supplementary Information (SI)), it seems to me that a small water droplet with volume $<10 \mu\text{L}$ should float on the ferrofluid that is used in the study (see the paper Phan, Stability of a Floating Water Droplet on an Oil Surface, *Langmuir* 2014, 30, 768–773). If this is possible, the freezing delay should be measured by placing 'small' drops on the ferrofluid – without applying any magnetic field. The freezing delay should be maximal in this case and will offer an independent validation for the authors' results. On the other hand, if the authors think that the magnetic field itself has a role in freezing delay – unlike the arguments made in the manuscript which suggest that the stronger field leads to lower peaks in Rosensweig instability and hence reduces the water droplet and substrate contact – then this should be clearly discussed. This clarification will also be pertinent to the comment 1.
3. The ferrofluid should have ferromagnetic nanoparticles in it, thus I am left questioning the assumption of molecularly smooth interface that runs through the work. The authors should characterise their ferrofluid by analysing nanoparticle size and its distribution, particle concentration and the carrier solvent. This should be followed by an attempt to assess the role of particle size on the ice nucleation.
4. The calculation of 'm' is elegant. However, the analysis is left incomplete. The authors should compute the freezing delay and/or nucleation temperature (as in ref. 26) to get a meaningful assessment of how close their estimate is to experiments.
5. The exact methodology of adhesion measurement needs to be stated. To this end, I would also recommend that the authors use a recent work "Tuteja and co-workers, Designing durable icephobic surfaces, *Science Advances* 2016, 2(3), e1501496" to help explain the ultra-low adhesion measured on their surfaces.
6. It is unclear whether the temperature inside the experimental chamber was uniform. As far as I can tell from the somewhat unclear description in the SI (section 5), the

nitrogen is used for insulation, whereas the cold plate helps cooled the samples under test. The humidity of the chamber should also be measured.

Minor comments:

7. Clarification: Figure 1c, bottom-most panel with three images of droplet inside the ferrofluid: are these schematics or black and white images from the camera.
8. How were the error bars in Figure 2 calculated? Please include a small section in the Supplementary Information (SI) section for this.
9. Line 150, page 9: the unit should be 'mm'?
10. Inset in Figure 4c: Is it a cartoon or an experimental image? If latter, the contact angle appears too high. Was the ice substrate smooth or rough?
11. Rather than calling it 'stability criterion' for depletion rate, I would say 'assumed critical threshold' for depletion rate. If this corresponds to a certain lifetime, based on the thickness of the ferrofluid film used, please specify that.

In summary, this paper contains some very exciting experimental results. Upon addressing the above comments, I will fully support its publication.

REVIEWERS' COMMENTS:

Reviewer #1 (Remarks to the Author):

The authors have toned down their claims and have added in sufficient repeats that I am satisfied that the findings can be published now.

I have no doubt that the need for magnetic fields will preclude the use in most situations in real life - but the idea and concept are worthy of publishing.

Reviewer #2 (Remarks to the Author):

I read the extensive responses of the authors to my comments and appreciate the fact that they did significant additional work based on them to improve the manuscript. The manuscript is now much improved and I recommend it for publication.

Revision Report

Reviewer #1:

The idea of icephobicity using magnetic surfaces is both novel and fascinating. The results look encouraging indeed and as such should be published sooner rather than later.

1) *“The measurement of lag times (time until nucleation for supercooled droplets) is probably not the best way to proceed however and some further measurements are required with multiple measurements on each sample.”*

Author reply: We agree with the reviewer that more experiments will enrich our manuscript. Initially, we had conducted ten experiments for each data point, but in the revised version we have included new experiments. The lag time criterion was chosen based on the paper of Eberle et al.¹ “Rational nano-structuring of surfaces for extraordinary icephobicity.” This work reports time lags for ice nucleation on nano and hierarchical structures. This criterion allows us to directly compare our results on MAGSS with other state-of-the-art icephobic surfaces. We emphasize that in addition to lag time, we used median nucleation temperature and adhesion strength as other criteria for comparison.

We have conducted new experiments: 44 experiments at -34 °C, 24 experiments at -30 °C, and 19 experiments at temperature of -25 °C. All of these measurements and their probabilities are shown in Supplementary Figures 11-13. Furthermore, for measurements of lag time as a function of the magnetic field, the new experiments are tabulated in the Supplementary Table 2. Based on these new measurements, we have modified Fig. 2b and Fig. 2c in the manuscript.

Action: In the supplementary information, we have included the new lag time measurements, Figs. 11-13 and Table 2.

2) *“I have no doubt the conclusions are valid - but they have not been sufficiently proved yet - it is well known that a given liquid sample if supercooled may freeze after say 200 seconds but on the next run may not freeze for 10000 seconds. This stochasticity seems not to have been discussed in that aspect of the measurements.”*

Author reply: The role of heating/cooling cycles on the measured values of T_N and τ_{av} is studied. In these cyclic experiments, the MAGSS was placed in the icing test chamber and water droplets were deposited on MAGSS. The temperature of MAGSS is reduced in a quasi-steady approach and T_N and τ_{av} are measured. Once all of the droplets were frozen, MAGSS was heated to 5 °C to form water droplets again. MAGSS was cooled again to low temperatures and T_N and τ_{av} were measured. We conducted five cycles of heating and cooling. The change in T_N is ± 1 °C and the change in τ_{av} was ± 10 s. The extreme icephobicity of MAGSS is unaffected by heating/cooling cycles.

Action: We have conducted new experiments on the role of heating/cooling cycles on T_N and τ_{av} values. We have reported all these experiments in the Supplementary Fig. 17, Supplementary Note 7.

3) “ p1,118 - normally do not have acronyms in abstracts ”

Author reply: Thank you. We have corrected this mistake.

Action: We have removed the acronyms in the abstract.

4) “ p2,32,33 the English needs fixing, i.e. longevity due to ... ”

Author reply: We have edited the entire manuscript and have corrected the grammatical errors.

Action: The manuscript and supplementary information were edited thoroughly.

5) “ p2, 138 don’t use paradigm twice in same paragraph ”

Author reply: We have corrected the sentence. The new sentence reads “Here, we report a new paradigm and corresponding surface with extreme icephobicity, high mobility for liquid and ice, self-healing, and stability at high Reynolds numbers. In this new approach, magnetic liquid-liquid interfaces are exploited to achieve these unprecedented characteristics.”

Action: The paragraph was edited.

6) “ p3, 153 delete the before hetero... ”

Author reply: We corrected the grammatical errors.

Action: The manuscript was thoroughly edited.

7) “ p3 157 - explain better that photograph of the water becoming embedded - it is not clear when compared to slips coatings ”

Author reply: For SLIPS, we have studied the formation of the solid-water interface both mathematically and experimentally. In the Supplementary Note 5, the details of this thermodynamic analysis are provided. Once a water droplet sits on SLIPS, two thermodynamic states can occur: (I) coexistence of water-oil and solid-oil interfaces or (II) only solid-water interface. The thermodynamic analysis shows that configuration (II) is a stable configuration. To show partial formation of a solid-water interface on SLIPS, we compared the mobility of a water droplet on MAGSS and SLIPS in Supplementary Movie 2. Droplet mobility on MAGSS is much higher than SLIPS. Formation of the solid-water interface reduces the droplet mobility on SLIPS due to partial pinning. *In addition*, similar ice adhesion strength on SLIPS and nano-structured surfaces is another indication of the partial existence of the solid-water interface.

Action: We have added a new section to the supplementary information (Note 5) and have shown both mathematically and experimentally the partial formation of the solid-water interface in SLIPS.

8) “ p5, l2 more explanation needed of droplet shapes typically on superhydrophobic surfaces ”

Author reply: The droplet contact angle on the superhydrophobic surfaces is generally more than 150° . The schematic in Fig. 1 (a) is intended to show the high contact angle and the formation of a solid-water interface.

Action: The new sentence reads “For superhydrophobic surfaces, once a water droplet touches the surface, it forms a high-energy solid-water interface with a contact angle of $>150^\circ$.”

9) “ p5 l84 - promotes nucleation - no evidence and not generally what is found at all - delete or explain ”

Author reply: High-energy solid-water interface reduces the Gibbs energy barrier for ice nucleation, leading to ice formation at higher temperatures. We agree with the reviewer that we have not shown this fact in this work. Thus, we modified the sentence as “We studied the infusion of a water droplet into a finite liquid oil layer through a Hele-Shaw cell in Fig. 1c. In a Hele-Shaw cell with a thickness of 3 mm between two glass slides, we placed a water droplet with a volume of 30 μL on top of a black oil surface, which is in contact with a cold solid substrate at a temperature of -25°C . The black oil is intentionally used to visualize motion of the water droplet and establishment of a solid-water interface. This method allows us to observe the water-oil interface. Although the buoyancy force opposes infusion of the water droplet to the bulk oil, higher surface energy of the water-air interface compared to the oil-water interface pulls the water droplet to the bulk oil and forms a solid-water interface. Formation of this interface decreases the energy barrier of water-ice phase change and favors heterogeneous ice nucleation. The short ice nucleation time is an indication of the formation of a solid-liquid interface.”

Action: The sentence was modified.

10) “ p7 fig 2 b, need to know sample numbers for each test - if $n=1$ then need to be done 10 times as minimum ”

Author reply: In Fig. 2 (b), we conducted 44 experiments on τ_{av} at a temperature of -34°C , 24 experiments at a temperature of -30°C , and 19 experiments at a temperature of -25°C . We have included all of this data in Fig. S11-S13. We emphasize that as temperature increases, the ice nucleation delay time rises exponentially, and for each droplet at -25°C temperatures, we should wait for > 16 hrs for freezing. Thus, the number of conducted experiments is lower.

Action: We have conducted new experiments and all of the data and error bars are represented in Fig. 2b and in section of the Supplementary Note 7.

11) “ p7 l 124 - again $n=?$ ”

Author reply: For the Fig. 2c, we have conducted new experiments, as tabulated in Supplementary Table 2. The mean value and the error bar are represented in Fig. 2c.

Action: The figure is modified with the new data.

12) “ *p12 l203 delay change to drop or lower*”

Author reply: We changed it to lower. The manuscript is thoroughly edited.

Action: The text was edited.

We again thank the reviewer for the thoughtful comments.

Reviewer #2:

1) *This paper reports an experimental study on a new kind of icephobic surface with extreme performance, which the authors demonstrate with a series of experiments. There are a few things that from the outset seriously limit my enthusiasm about this work. The first is that this is an "active" surface. It requires a magnetic field the magnitude of which depends on application and surface size and as such it is inherently limited to ferrofluids that are responsive to this. Think of an airplane wing, a power transmission line, the surface of a ship or the blade of a large wind turbine with a liquid (ferrofluid) surface that is magnetically activated, as the authors mention at the end of the abstract. It is very difficult (actually impossible) to believe that these applications have in reality any relevance to the lab level surfaces presented in this paper.*

Author reply: We thank the reviewer for raising this point. Both SLIPS and MAGSS surfaces form a liquid-air interface rather than a solid-air interface. In SLIPS, the oil on the surface is maintained by a capillary force induced by the underneath porous solid, while for MAGSS, a permanent magnetic tape is placed underneath of the solid to keep the ferrofluid at the surface. Since the magnetic tape is **permanent** and change in its magnetic field is (< 1% over a hundred years), the MAGSS surfaces should be considered passive. That is, no external active actuation is needed to keep the icephobicity of these surfaces.

There are several advantages to these surfaces compared to SLIPS as listed below:

- (1) They do not need any micro/nano-structuring to induce the capillary force and can be applied to any surface (smooth, rough, metals, ceramic, and polymers). In other words, they suppress the role of the solid in the icephobicity.
- (2) At high-shear flow applications, which occur in airplane wings, power transmission lines, arctic vessels, and wind turbines, the airflow or water flow easily removes oil on SLIPS surface (see Movie for review) and the solid surface emerges. However, in MAGSS, the induced magnetic force locks the ferrofluid layer at the surface. We have shown this stability in Supplementary Movie 9 and 10. We ran these experiments for more than 15 hrs.
- (3) As shown by our analytical and experimental study, once a water droplet sits on SLIPS, it forms a solid-water interface that reduces the energy barrier for ice formation leading to higher values of T_N (Supplementary Note 5). However, for MAGSS, the induced volumetric force prevents the formation of a solid-water interface leading to extreme icephobicity. To show partial formation of a solid-water interface on SLIPS, we compared the mobility of a water droplet on MAGSS and SLIPS in Supplementary Movie 2. Droplet mobility on MAGSS is much higher than SLIPS. Formation of the solid-water interface reduces the droplet mobility on SLIPS due to partial pinning. *In addition*, similar ice adhesion strength on SLIPS and nano-structured surfaces is another indication of the partial existence of the solid-water interface.
- (4) The measured ice adhesion on SLIPS is in the order of other superhydrophobic surfaces (100 kPa), while this adhesion is in the range of 2-3 Pa for MAGSS. This low ice adhesion strength is a consequence of formation of the ice-ferrofluid interface. We

emphasize that even at extremely low temperatures, once ice forms on MAGSS, it slides from the surface by a minimal force preventing ice accretion on the surface.

- (5) The oil used in SLIPS is a very expensive fluid, while in MAGSS ferrofluids with a variety of oils can be used for development of these surfaces. The cost of ferrofluid used in this study is less than 50 cents per m². Also, the cost of permanent magnetic tape used in this study is < \$5 per m².

Action: We have added a new section to the Supplementary information, (Note 5) to show the exclusion of solid-water interface in MAGSS. Also, this effect is shown in supplementary Movie 1.

2) *“Then the authors expend a serious effort trying to claim that their surface is a new paradigm and not the same with SLIPS. I disagree. They rely on the same principle (slipping on a liquid layer), with the additional very serious restriction that their surface needs to be magnetically activated. SLIPS are passive surfaces—a huge advantage in my opinion (and we still have to see any significant applications even with SLIPS). In the original SLIPS papers (the one by Aizenberg and co-workers, for example) actually the droplet was simply riding on a film of liquid (not sinking through it), the porous texture serving the stability of the film. In this respect the claim that the magnetic liquid needs not texture (also vis. claimed advantage of no contact with the solid of the supercooled water hence ice nucleation inhibition) is not proven/correct.”*

Author reply: We emphasize that the principles of MAGSS and SLIPS are distinct. The MAGSS surface is a new paradigm in which we use volumetric force in the fluid to keep the liquid at the solid surface, while for SLIPS micro/nano-structuring is required for the introduction of capillary force. As discussed, since the magnetic tape is **permanent** and change in its magnetic field is < 1% over a hundred years, the MAGSS surfaces should be considered passive. That is, no external active actuation is needed to keep icephobicity of these surfaces.

We have shown both mathematically and experimentally that the solid-water interface forms in SLIPS, which is the reason for its high value of T_N and high ice adhesion strength. We have added a new section in the supplementary information (Note 5) and supplementary Movie 1 and 2.

Action: A section has been added to the supplementary information to show the formation of solid-water interface in SLIPS. Furthermore, we developed SLIPS surfaces exactly as procedure outlined in Wong et al.². We conducted the icephobicity measurement on SLIPS in our icing test chamber in same experimental conditions (Supplementary Figs. 15-16). The measured value of T_N for SLIPS is -25 ± 1 °C while this value for MAGSS is -34 ± 1 °C.

3) *“Once we allow for fields, electrical, magnetic etc to impart the surface functionality, then it is much easier to reach impressive numbers (as in this paper with very high supercooling and very low adhesion) but then the real problem is that these surfaces remain at the impressive “toy” level with no chance to real applications.”*

Author reply: As stated above, MAGSS functions with a **permanent** magnetic field. No external active actuation is needed to keep icephobicity of these surfaces. Thus, these surfaces should be considered passive.

4) “ *So, I find it a lot more appropriate to say that the MAGSS surfaces of this paper are as special case of a limited active SLIPS surface, rather than they are a new paradigm and all these other superlatives the authors use for their surface (overselling). This is what I think is the real perspective on the importance and novelty of this work.* ”

Author Reply: We have provided all the theoretical and experimental support to show the extreme icephobicity of MAGSS. These icephobic surfaces show ice formation temperature of -34 °C, 2-3 orders of magnitude higher delay time in ice formation, **extremely low** ice adhesion strength (≈ 2 Pa), and stability under shear flows up to Reynolds number of 10^5 . We have shown that these inexpensive surfaces are universal and can be applied to all types of solids (no required micro/nano structuring) with no compromise in their unprecedented properties.

5) “ *In Figure 1 the explanations on possible heterogeneous nucleation of the droplet on a SLIP surface are purely speculative and indeed self-serving. No experiments whatsoever were performed on correctly fabricated and optimized SLIP surfaces for this purpose and no measurements were made of the corresponding T_N and τ_{av} to prove a real comparison, despite the big claims of the authors. Results of other studies for other materials and conditions are shown for comparison, but this is far from conclusive. These other surfaces are all passive and no attempt was made (for example in the SLIP) surfaces to optimize with respect to nucleation.* ”

Author Reply: We have added a section to the supplementary information (Note 5) to show both analytically and experimentally that a solid-water interface forms on SLIPS leading to high T_N . Furthermore, we developed SLIPS surfaces in exactly same fashion as the procedure outlined in Wong et al.². Once SLIPS were developed, we conducted the icephobicity measurement on SLIPS in the icing test chamber. The measured value of T_N for SLIPS is -25 ± 1 °C while this value for MAGSS is -34 ± 1 °C. These results are presented in the Supplementary Figs. 15-16. The value of T_N measured for SLIPS is the same as the previously reported value.

Action: We have added a new section to the supplementary information. In this section, we have shown both analytically and experimentally that a solid-water interface forms on SLIPS, leading to a high T_N and high ice adhesion strength.

6) “ *Regarding the same measurements with the magnetic surfaces of this paper: Little details are given in the SI. These measurements are tricky and require statistics, usually from observing many drops simultaneously and determining freezing (or condensation, it is similar) probabilities with defined error ranges. None of this is done.* ”

Author Reply: We emphasized that the reported values are average value and not for one experiment. It was for 10 measurements. However, we have conducted more measurements in the revised manuscript. For T_N measurements, we have conducted 35 experiments. All the

measured data is reported in the Supplementary Fig. 10. We have also reported the probability density of our measurements. For Fig. 2 b, we conducted 44 experiments on τ_{av} at a temperature of -34 °C, 24 experiments at a temperature of -30 °C, and 19 experiments at a temperature of -25 °C. We emphasize that as temperature increases, the ice nucleation delay rises exponentially and for each droplet at -25 °C, we should wait for > 16 hrs for freezing. Thus, the number of conducted experiments is lower. All these measurements are shown in Supplementary Figs. 11-13.

Action: We have conducted new experiments for T_N and lag time and the new data is represented in Supplementary Fig. 10-13.

7) Also what is the effect of environmental conditions in the chamber? One has to also monitor humidity to define the thermodynamic state. What is this effect? None of this is done/explained. Note that due to exponential behaviors such measurements are very sensitive. I am not convinced of the correctness of the reported results."

Author Reply: We monitored the thermodynamic state of the chamber in our experiments, Fig S9. These details are provided in the supplementary information, section 7. The temperature of the icing test chamber was measured at 10 points and the deviation in the temperature was $\pm 1^\circ\text{C}$. The humidity of the test chamber was measured with a humidity sensor and was $80 \pm 10\%$. The role of evaporation on the ice nucleation is discussed by Dr. Poulidakos' group in "Mechanism of supercooled droplet freezing on surfaces, Nature Communication, 2012, 3, 615". In summary, for a sessile droplet, evaporation at the liquid-air interface causes local cooling of the liquid resulting into homogenous nucleation of ice.

Action: The condition of the icing test chamber is shown in Supplementary Fig. 9.

8) "No thermodynamic analysis has been made to come up with quantitative criteria on surface energies of various configurations and enable a scientific conclusion, other than observation from experiments. This should have been a must for the authors to establish the criteria under which the so-called volumetric magnetic force overcomes the surface energy differences they claim."

Author Reply: We have conducted a thorough thermodynamic analysis to show the role of magnetic field in MAGSS. This information is presented in the supplementary information, section 5. The volumetric magnetic force excludes the role of the solid-water interface in ice nucleation. Also, supplementary video 1 shows this fact.

Action: We have added a new section in the supplementary information (Note 5) and have provided Supplementary Movies 1 to show the role of the magnetic field in MAGSS.

9) "No analysis has been performed to quantify the obviously formed surface waves or non uniformities on the surface of the ferromagnetic liquid, their dependence on the various process parameters and their effect on the entire process. A value of $B=347\text{ mT}$ is mentioned for the specific experiments performed in this paper after which the solid water interface is excluded

and the nucleation time remains constant. This is purely empirical and specific. One needs a thorough general analysis for useful generic conclusions.”

Author Reply: In the new supplementary information, we have provided a thermodynamic analysis (Note 5) to show the role of magnetic force in exclusion of the solid-water interface. Through this analysis, we found that magnetic field in range of 300 mT is required to completely exclude the solid-water interface. The provided thermodynamic platform is general and can be applied to any solid-ferrofluid combinations. The surface waves form on the MAGSS due to competition between the magnetic force and the surface tension force. Currently, there is no theoretical model for the wavelength of these curves as discussed in Supplementary Note 4. We know that the observed surface waves are a competition between the surface tension force and the magnetic volumetric force. As we increase magnetic volumetric force (or reduce surface tension of the ferrofluid), the wavelength and amplitude of these waves can be reduced. In Supplementary Fig. 4, the change in the amplitude of the surface waves as a function of the induced magnetic field is shown.

Action: A new section is added to the supplementary information (Note 5) to show quantitatively how a threshold magnetic field is required to exclude solid-water interface. In the Supplementary Note 4, we have discussed the appeared surface waves on the ferrofluid interface.

10) *“The adhesion test results are impressive but again, this is a magnetically activated effect with little bearing to real applications in technology nor manifestation in nature. Is it at all relevant to compare these results to passive surfaces (Lotus-based, or SLIP?).”*

Author Reply: As we discussed above, the MAGSS functions with **permanent** magnetic field. No external active actuation is needed to keep icephobicity of these surfaces. Thus, these surfaces should be considered passive and they will remain icephobic in a long-time operation.

We did not claim any manifestation of these surfaces in nature. We have compared the figures of merit for icephobicity of these surfaces with other state-of-the-art icephobic surfaces and have shown superior characteristics of these surfaces.

11) *“Durability tests and videos are shown for cases showing robustness for about 30 min. In what real application mentioned by the authors (turbines, boats, powerlines and aviation) is this nearly enough?”*

Author Reply: We agree with the reviewer. We have conducted new experiments and tested the MAGSS sample under flow of water and air for more than 15 hrs. These experiments are provided in Supplementary Movies 9 and 10. No change in the characteristics of MAGSS was observed after long-run tests.

Action: New experiments are conducted to show durability of the MAGSS under shear flow of water and air. These new results are presented in Supplementary Movies 9 and 10.

12) *And how are the high magnetic fields to be applied to ensure durability at high Weber, and Reynolds numbers? The reported We numbers for example are low.*

Author Reply: The magnetic field is introduced through magnetic tapes. Magnetic tapes with high magnetic fields can be obtained to serve for this purpose. Also, the viscosity of ferrofluid can be tuned for application of high Weber numbers. We developed MAGSS with high viscosity and examined its self-healing characteristics at a Weber number of 600. The result is presented in Supplementary Movie 7, in which no change in surface characteristics of MAGSS is observed.

Action: We conducted new experiments at higher Weber numbers to show the self-healing characteristics of MAGSS.

13) Finally how about mechanical tests? Resistance to scraping off of the ferrofluid that removes the entire functionality? What fields does this need as a function of the scraping force?"

Author Reply: We have conducted a new experiment to show resistance of the MAGSS to scraping by a sharp object. This new experiment is shown in Supplementary Movie 8. As shown in the video, no change in the surface characteristics of MAGSS is observed after the scarping test.

Action: We have conducted a new experiment on resistance to scraping and the result is presented in Supplementary Movie 8.

We again thank the reviewer for the thoughtful comments.

Reviewer #3:

“The paper by Irajizad et al. reports a new way to make icephobic surfaces using magnetically energized ferrofluid on surfaces. It is a creative and simple approach and there are some exciting experimental results, which will be of broad interest. I am inclined to support the publication of the work, but there some major scientific aspects that need to be clarified.”

1) *“The physics of freezing delay on MAGSS is not clear and needs more work. Vital to the working of these icephobic surfaces is the stability of a water droplet on the field activated ferrofluid puddle (with Rosensweig projections). Under no magnetic field the water droplet sinks into the ferrofluid, which ruins the icephobicity. Therefore, it is critical to understand this stability. Simply making a broad statement that with magnetic field the body forces will start to dominate the interfacial forces (as the authors do on the page 6 of the manuscript, “In the presence of ...”) does not address this. Freezing delay is an interfacial surface energy dependent phenomenon. Why should the change in body force - in the ferrofluid – influence stability of a water droplet sitting on it or the freezing of the water droplet? The authors will need to assess the rheology of the ferrofluid under magnetic field and then use a force balance to help substantiate their results and explain them.*

Author Reply: We thank the reviewer for this insightful comment. We have added a new section to the supplementary information, (Note 5). In this section, we have studied the thermodynamic stability of a water droplet on an oil surface. The stable thermodynamic state requires sinking of the droplet in oil and formation of a solid-water interface. Next, we have included the role of the magnetic field in the stability of a water droplet on a ferrofluid surface. We have shown that a threshold magnetic field is required to exclude formation of a solid-water interface.

Action: We have added a new section to the supplementary information, (Note 5), and studied the stability of a water droplet on a ferrofluid surface.

2) *“Please see the works by “Phan et al. Can Water Float on Oil? Langmuir 2012, 28, 4609–4613.” It offers insight on interfacial force balance and droplet stability. Looking at the interfacial tension values measured (section 7 of the Supplementary Information (SI)), it seems to me that a small water droplet with volume $<10 \mu\text{L}$ should float on the ferrofluid that is used in the study (see the paper Phan, Stability of a Floating Water Droplet on an Oil Surface, Langmuir 2014, 30, 768–773). If this is possible, the freezing delay should be measured by placing ‘small’ drops on the ferrofluid – without applying any magnetic field. The freezing delay should be maximal in this case and will offer an independent validation for the authors’ results. On the other hand, if the authors think that the magnetic field itself has a role in freezing delay – unlike the arguments made in the manuscript which suggest that the stronger field leads to lower peaks in Rosensweig instability and hence reduces the water droplet and substrate contact – then this should be clearly discussed. This clarification will also be pertinent to the comment 1.”*

Author Reply: We thank the reviewer for pointing us to these papers. In two articles (Phan et al. Langmuir, 2012, 28, 4609-4613 and Langmuir, 2014, 30, 768-773), the stability of a water droplet on an oil surface is investigated. The authors have shown that the induced downward force due to density difference can be balanced through interfacial forces. Thus, a water droplet can be hanged at the oil interface without sinking. The upward interfacial forces by surface tensions overcome the downward gravitational forces to keep the droplet suspended at the oil surface. In the 2012 paper, the authors used a combination of water and oil with surface tensions of 44.9 and 21.7 mN/m, respectively. They showed that the water droplets with obtuse contact angles (~120) are stable at the interface. In the 2014 paper, the authors reduced the surface tension of water to 30.0 by surfactants and used oil with surface tension of 8 mN/m and concluded that a water droplet can be stable with an acute contact angle (<90) as well.

There is one fundamental difference between our study and these studies. In our study, we have a solid surface present, which forms an interface with oil. Furthermore, the surface tension of DI water in our study is 72 mN/m and the surface tension of oil is 24 mN/m. Thus, we could not directly compare our work with the works of Phan et al. However, we took two approaches to show that the role of the magnetic field is essential in MAGSS.

Initially, we analytically studied the thermodynamic stability of a water droplet on an oil layer on a solid surface. This new section is added to the supplementary information, (Note 5). The analysis shows that the water droplet replaces the oil layer in the stable thermodynamic state. In the next approach, we conducted an experimental study in which droplets in the volume range of 4-180 μ L were deposited on a ferrofluid layer with and without magnetic field as shown in Supplementary Fig. 18. In the absence of a magnetic field, water sinks in the ferrofluid layer and forms a solid-water interface. However, in the presence of a magnetic field, this interface is excluded. This fact is also shown in Supplementary Movie 1.

Action: We have added a sentence in the manuscript to include the new references. The sentence reads “Infusion of a water droplet into a liquid oil bath was studied by Phan et al. The studied oil had lower density than that of water. They found that the downward gravitational force (induced by density difference) can be balanced by interfacial tension forces for certain combination of low surface tension water and oil. However, there was no solid surface in their study and thickness of the oil bath was infinite.”

Two new sections have been added to the supplementary information to discuss both analytically and experimentally the role of magnetic field on the extreme icephobicity of MAGSS.

3) *“The ferrofluid should have ferromagnetic nanoparticles in it, thus I am left questioning the assumption of molecularly smooth interface that runs through the work. The authors should characterize their ferrofluid by analyzing nanoparticle size and its distribution, particle concentration and the carrier solvent. This should be followed by an attempt to assess the role of particle size on the ice nucleation.”*

Author Reply: The single domain iron-oxide nano-particles in the ferrofluid are coated with surfactants to be suspended in the ferrofluid matrix. The used surfactants are proprietary information of CMS Magnetics and we could not obtain the information on the chemical formulas of these surfactants. To determine the dimensions of the iron-oxide nanoparticles, we

diluted the ferrofluid with acetone several times. A smooth Si substrate was prepared by plasma cleaning. A droplet of diluted ferrofluid was deposited on the Si substrate and the liquid was allowed to evaporate. A ring trace of the particles was formed on the surface due to coffee-ring effect. The sample was placed in the Scanning Probe Microscope (SPM, Bruker Multimedia 8) and the distribution of particle size was measured. We noted that aggregation of particles occurred in the evaporation process. However, the smallest particles were in the range of 10-20 nm as shown in Supplementary Fig. 3. We should emphasize that these nano-particles are covered with surfactant molecules. That is, once they float on the surface of the ferrofluid, the roughness of the ferrofluid surface is determined by the structure of surfactant molecules and consequently can be assumed the molecularly smooth. We agree with the reviewer that a study on role of particle size on icephobicity is interesting, but this study is out of scope of this manuscript. We are currently following this idea.

We measured the chemical composition of the ferrofluid through FTIR (Nicolet 4700 FTIR, Thermo Electron Corporation). These measurements are shown in Fig. S2. The oil mainly seems consist of a family of fatty acids, primarily Lauric acid, $\text{CH}_3(\text{CH}_2)_{10}\text{COOH}$. At wavenumber of 3000 cm^{-1} , peak of carboxylic acid group (-COOH) appears. The peak corresponding to the alkenyl group (R-CH=CH_2) appears around a wavenumber of 1100 cm^{-1} . Traces of other fatty acids such as Myristic and Palmitic acids can also be seen in the FTIR spectrum.

Action: We have included all these new analysis in the Supplementary Note 1.

4) *“The calculation of ‘m’ is elegant. However, the analysis is left incomplete. The authors should compute the freezing delay and/or nucleation temperature (as in ref. 26) to get a meaningful assessment of how close their estimate is to experiments.”*

Author Reply: We thank the reviewer for this thoughtful point that enriched our manuscript. Once we measured the value of m in MAGSS, we used this m value along with nucleation theory to determine the ice nucleation delay time, τ_{av} , and compared with the measured τ_{av} . The delay time is written as

$$\tau_{av} = \frac{1}{J(T)}$$

where $J(T)$ denote the rate of formation of critical ice embryo and is written as

$$J(T) = K(T)A \exp\left(\frac{-\Delta G^*(T)}{k_b T}\right)$$

where $K(T)$ denotes the kinetic factor for diffusion of water molecules across ice surface, A the water-solid contact area, ΔG^* Gibbs energy barrier for formation of critical ice embryo, and k_b Boltzmann constant. The value of m plays a role in ΔG^*

$$\Delta G^* = \frac{8\pi\gamma_{IW}^3}{3\Delta G_v^2} f(m, x)$$

Once we measured m value, we used the determined values of $K(T)$ and A to predict τ_{av} .

$$K(T) = 2.08366 \times 10^{29} T \exp\left(\frac{-892 T}{(T - 118)^2}\right)$$

The surface tension of the ice-water interface plays a significant role in these calculations. As discussed by Nemeč³, there is ± 1.1 mN/m uncertainty in this value. In Supplementary Fig. 21, we have compared our measurements with predictions. The measurements are within the uncertainty of the predictions. We should emphasize that in these calculations, we have accurately considered temperature-dependence of all parameters.

Action: We have added a new section to the manuscript to determine the time delay for nucleation through the measured value of m , Supplementary Note 9.

5) *“The exact methodology of adhesion measurement needs to be stated. To this end, I would also recommend that the authors use a recent work “Tuteja and co-workers, Designing durable icephobic surfaces, Science Advances 2016, 2(3), e1501496” to help explain the ultra-low adhesion measured on their surfaces.”*

Author Reply: We have elaborated on the approach for ice adhesion measurement. Since ice adhesion strength on MAGSS is so small, the induced gravitational force by titling can be used to determine ice adhesion strength on MAGSS. We placed the MAGSS sample in the icing test chamber and deposited a water droplet on the surface. The temperature of the test chamber is reduced to -34 °C to form ice. Next, the test chamber was tilted stepwise with a 1° tilt in each step. The threshold angle for the sliding of ice on the surface was measured. Through measured threshold tilt angle, volume of ice and the contact area of ice, we determined the adhesion strength of ice on MAGSS.

We have included Golovin et al.⁴ as a reference in our manuscript. The sentence in the manuscript reads “Golovin et al. utilized interfacial slippage to achieve icephobic surfaces with a ice adhesion strength of 200 Pa. In their approach, they introduced uncross-linked polymeric chains on the surface to achieve no-slip boundary condition at the solid-ice interface.”

Since the ice adhesion strength in Golovin et al. is high, they use the developed approach by Meuler et al.⁵ for ice adhesion measurements. In this approach, pillars of ice are formed on the icephobic surface. A force transducer is utilized to apply force on the side of pillars, resulting in shear stress at the solid-ice interface. The threshold force of the slide of pillars is used to determine the ice adhesion strength. Also, We emphasize that even at extremely low temperatures, once ice forms on MAGSS, it slides from the surface by a minimal force preventing ice accretion on the surface

Action: We have included the work of Golovin et al. in our manuscript. We added included detail of ice adhesion measurements in the Supplementary Note 7.

6) *“It is unclear whether the temperature inside the experimental chamber was uniform. As far as I can tell from the somewhat unclear description in the SI (section 5), the nitrogen is used for insulation, whereas the cold plate helps cooled the samples under test. The humidity of the chamber should also be measured.”*

Author Reply: We measured the temperature of the test chamber at 10 coordinates on top of the cold plate. The measured temperatures are shown in Supplementary Fig. 9. The temperature fluctuation in the test chamber is ± 1 °C. The humidity of the test chamber during the experiments was measured with a humidity sensor. The humidity is 80 ± 10 %.

Action: This information is provided in Supplementary Note 7.

7) *“Clarification: Figure 1c, bottom-most panel with three images of droplet inside the ferrofluid: are these schematics or black and white images from the camera.”*

Author Reply: This is a real experiment in the Hele-Shaw configuration. We studied the infusion of a water droplet into a finite liquid oil layer through a Hele-Shaw cell in Fig. 1c. In a Hele-Shaw cell with a thickness of 3 mm between two glass slides, we placed a water droplet with a volume of 30 μL on top of a black oil surface, which is in contact with a cold solid substrate at a temperature of -25 °C. The black oil is intentionally used to visualize motion of the water droplet and establishment of a solid-water interface. This method allows us to observe the water-oil interface. Although the buoyancy force opposes infusion of the water droplet to the bulk oil, higher surface energy of the water-air interface compared to the oil-water interface pulls the water droplet to the bulk oil and forms a solid-water interface. Formation of this interface decreases the energy barrier of water-ice phase change and favors heterogeneous ice nucleation. The short ice nucleation time is an indication of the formation of a solid-liquid interface.

Action: In the caption of Fig. 1, the following sentence is added to avoid confusion: “In a Hele-Shaw configuration, displacement of the water droplet to the surface in the presence of a magnetic field is shown. That is, the volumetric force by the magnetic field does not allow the formation of a solid-water interface.”

8) *“How were the error bars in Figure 2 calculated? Please include a small section in the Supplementary Information (SI) section for this.”*

Author Reply: We have provided a new complete section in the supplementary information on the number of measurements for each point and the probability density of the measured data. This is shown in the Supplementary Figs. 10-13 and Supplementary Table 2.

Action: A new section is added to the supplementary information on the number of measurements for T_N and τ_{av} .

9) *“Line 150, page 9: the unit should be ‘mm’?”*

Author Reply: Thank you. We corrected the mistake.

Action: We changed the unit to “mm”.

10) *“Inset in Figure 4c: Is it a cartoon or an experimental image? If latter, the contact angle*

appears too high. Was the ice substrate smooth or rough?”

Author Reply: This is a picture of a real experiment. In these experiments, we measured the contact angle of ferrofluid on smooth ice in a water medium. This allows us to calculate the value of m . A glass beaker, thoroughly cleaned, was prepared. 10 mL of DI water was poured in the beaker and the beaker was placed on a cooling stage to form into ice. The ice started to grow from the bottom of the beaker until the whole volume of water transformed to ice. The surface of the ice was completely smooth with an optical microscope. However, the ice surface may have some roughness at the nano-scale depending on the growth mechanism at ice-water interface (Dendritic or flat solidification). Then, a ferrofluid droplet was deposited on the ice surface and the beaker was filled with water in a quasi-steady manner to avoid any further ice growth. The contact angle of the ferrofluid droplet on ice was measured once the introduced water completely covered the ferrofluid droplet. Note that the temperature was measured at the ferrofluid-ice interface.

Action: This description is added to the Supplementary Note 9.

11) *“Rather than calling it ‘stability criterion’ for depletion rate, I would say ‘assumed critical threshold’ for depletion rate. If this corresponds to a certain lifetime, based on the thickness of the ferrofluid film used, please specify that.”*

Author Reply: We changed it to “assumed critical threshold”. We had defined this threshold in the caption of Fig. 3. It is less than 2 $\mu\text{m/hr}$. This is the limit of our optical imaging technique.

Action: We changed the “stability criterion” to “assumed critical threshold”.

“In summary, this paper contains some very exciting experimental results. Upon addressing the above comments, I will fully support its publication.”

We again thank the reviewer for the thoughtful comments.

References

- 1 P. Eberle, M. K. Tiwari, T. Maitra and D. Poulikakos, *Nanoscale*, 2014, **6**, 4874–81.
- 2 T.-S. Wong, S. H. Kang, S. K. Y. Tang, E. J. Smythe, B. D. Hatton, A. Grinthal and J. Aizenberg, *Nature*, 2011, **477**, 443–447.
- 3 T. Nemeec, *Chem. Phys. Lett.*, 2013, **583**, 64–68.
- 4 K. Golovin, S. P. R. Kobaku, D. H. Lee, E. T. DiLoreto, J. M. Mabry and A. Tuteja, *Sci. Adv.*, 2016, **2**, 1–12.
- 5 A. J. Meuler, J. D. Smith, K. K. Varanasi, J. M. Mabry, G. H. McKinley and R. E. Cohen, *ACS Appl. Mater. Interfaces*, 2010, **2**, 3100–10.